Lowland tapir distribution and habitat loss in South America

Cordeiro Jose Luis Passos jlpcordeiro@gmail.com zeluis@fiocruz.br 1
Fragoso José M.V. 2
Crawshaw Danielle 1
Oliveira Luiz Flamarion B. 3
1 Fiocruz Mata Atlântica, Fundação Oswaldo Cruz (Fiocruz) , Rio de Janeiro , Brasil
2 Department of Biology, Stanford University , Stanford , CA , United States
3 Museu Nacional, Departamento de Vertebrados, Universidade Federal do Rio de Janeiro , Rio de Janeiro , Brasil
Gandini Patricia
Electronic publication date: 2016 Sep 13
Publication date: 2016
Volume: 4
Electronic Location ID: e2456
Received 2015 Dec 9; Accepted 2016 Aug 17
Copyright: ©2016 Cordeiro et al.
Copyright year: 2016
Copyright holder: Cordeiro et al.
License: This is an open access article distributed under the terms of the Creative Commons Attribution License, which permits unrestricted use, distribution, reproduction and adaptation in any medium and for any purpose provided that it is properly attributed. For attribution, the original author(s), title, publication source (PeerJ) and either DOI or URL of the article must be cited.
License URL: https://creativecommons.org/licenses/by/4.0/

Keywords: Tapirus terrestris, Species distribution models, MAXENT, ENFA, Conservation planning, Protected areas

Funding: CNPq (Brazilian Research Council) A research grant to Oliveira LFB from CNPq, the Brazilian Research Council, provided funding for the completion of this work. The funders had no role in study design, data collection and analysis, decision to publish, or preparation of the manuscript.

==============================
The development of species distribution models (SDMs) can help conservation efforts by generating potential distributions and identifying areas of high environmental suitability for protection. Our study presents a distribution and habitat map for lowland tapir in South America. We also describe the potential habitat suitability of various geographical regions and habitat loss, inside and outside of protected areas network. Two different SDM approaches, MAXENT and ENFA, produced relative different Habitat Suitability Maps for the lowland tapir. While MAXENT was efficient at identifying areas as suitable or unsuitable, it was less efficient (when compared to the results by ENFA) at identifying the gradient of habitat suitability. MAXENT is a more multifaceted technique that establishes more complex relationships between dependent and independent variables. Our results demonstrate that for at least one species, the lowland tapir, the use of a simple consensual approach (average of ENFA and MAXENT models outputs) better reflected its current distribution patterns. The Brazilian ecoregions have the highest habitat loss for the tapir. Cerrado and Atlantic Forest account for nearly half (48.19%) of the total area lost. The Amazon region contains the largest area under protection, and the most extensive remaining habitat for the tapir, but also showed high levels of habitat loss outside protected areas, which increases the importance of support for proper management.

Introduction

The lowland tapir (Tapirus terrestris) is the largest terrestrial vertebrate (autochthone) in its ecosystems. Considered a keystone species, due to its large size and biomass, and also due to its function as seed predator/disperser (Bodmer, 1991; Rodrigues, Olmos & Galetti, 1993; Fragoso, 1997; Fragoso, 2005; Taber et al., 2009; Medici, 2010). Tapirs inhabit a variety of habitats, from xeric formations such as the Gran Chaco, to tropical dry forests and wetter formations such as rain forests, gallery forest, shrub forests, savannas and grasslands (Nowak, 1991; Fragoso & Huffman, 2000). These vegetation types, however, are used unevenly, with tapirs exhibiting selective habitat use. For example, they seem to prefer areas with moist palm forests, and wet, or seasonally inundated areas (Brooks, Bodmer & Matola, 1997; Fragoso & Huffman, 2000; Tobler, 2008; García et al., 2012).

The lowland tapir (Tapirus terrestris) maintains the most extensive distribution of the four recognized extant tapir species and inhabits the subtropical to tropical zones of South America, from northern Argentina, through Brazil, Bolivia, Peru, Ecuador, Venezuela, Guyana, Suriname, French Guiana and Colombia, east of the Atrato River (Nowak, 1991; Brooks, Bodmer & Matola, 1997; Groves & Grubb, 2011; Tirira, 2007; Wallace, Ayala & Viscarra, 2012). A fifth tapir species, still under discussion, was recently described (Cozzuol et al., 2013; Voss, Helgen & Jansa, 2014).

Taber et al. (2009) provides the most updated and detailed evaluation of T. terrestris distribution and conservation status. The authors estimate, based on specialists opinions and occurrence records, that historic distribution covered 13.129.874 km2 and the current distribution is 11.232.018 km2. T. terrestris is considered to be Vulnerable due to habitat loss, illegal hunting and competition with livestock. Most of the main habitat out of Amazon has been converted to human use as cattle ranching and agriculture in a short time. The species is completed absent in vast areas of its historic range (Naveda et al., 2008; Taber et al., 2009). Deforestation and other forms of habitat change have all contributed to population declines. Therefore, the understanding the role of variables associated with the original distribution patterns are crucial to partitioning factors involved in the viability of populations. Accordingly, large-scale assessments may show patterns which locally are not evident, but involved in the viability of populations and, to a great extent, the impacts of changes in the long-term.

Species occurrence may be related to set of predictors ranging from site to landscape scale, as range of natural vegetation, terrain attributes, disturbance and human scenarios as land use and protected areas, and other environmental variables as those characterizing climate and seasonal changes (Franklin, 2009; Peterson et al., 2011). Such comprehensive ecological evaluation, including species responses to global changes may be effective when incorporating a large area perspective, particularly in the tropics where data deficient species are the rule or vast areas have been transformed without adequate inventories.

Tapir, despite being a large mammal in the context of the Neotropics, is still data deficient in the largest data set collected for a species with wide distribution (Taber et al., 2009). However, the available data allows insights on their response to ecological factors along the eco-geographical regions and major habitats, and can support conservation planning showing patterns of response to ecological and human factors in the time scale.

Identifying the most important environmental parameters bounding species distributions remains difficult because animals respond to the environment at a range of spatial scales (Krausman, 1997). Ungulates for example, make foraging decisions both within and across a variety of spatial scales, making it difficult to relate species to specific habitats across their entire species range (Hobbs, 2003). However, describing these relationships is an important first-step towards understanding linked ecological processes and guiding conservation decision-making, as the agents that determine population viability may include factors related to habitat or elements that transcend spatial scales, such as dynamically linked variables or unlinked elements (Peterson et al., 2011). Species Distribution Models (SDM) are thus important tools for defining testable hypotheses and generating potential species’ ranges. Clements et al. (2012) and Mendoza et al. (2013) produced a SDM for Asian tapir (Acrododia indica) and Baird’s tapir (Tapirella bairdii), respectively, and demonstrated the applicability of SDM use in the evaluation and development of tapir conservation strategies. Darren (2014) applied SDM to understand the distribution of lowland tapir in a fragment of Atlantic forest in southeast Brazil and highlight the importance of a fundamental understanding of a species’ natural history to determine not only appropriate model parameters, but also the biological relevance of SDMs.

Appropriate model selection is critical when ecological as well as distribution oriented hypotheses are to be tested. The selection of an SDM should consider the theoretical underpinnings and practical applicability of the model as well as the hypothesis of interest (Jiménez-valverde, Lobo & Hortal, 2008; Kamino, Stehmann & Amaral, 2011). Ecological Niche Factor Analysis (ENFA) and MAXENT are two approaches that are presently used for describing distributions and classifying landscape suitability for species (Braunisch & Suchant, 2010; Rebelo & Jones, 2010; Rodríguez-Soto et al., 2011).

ENFA generates species distributions based on Hutchinson’s concept of the ecological niche by comparing known species locations and associated environmental variables to areas without locations but with the same environmental conditions (Hirzel et al., 2002). In contrast, MAXENT’S theoretical underpinnings are based on the maximum entropy principle and mathematically similar to a Poisson regression model (Renner & Warton, 2013). We modeled the potential distribution of the lowland tapir in South America using both methods and evaluated their relative accuracy.

The objective of this study was to describe habitat suitability, potential distribution and quantification of habitat loss (total and per ecoregions) for T. terrestris over its entire range, to evaluate and contribute to the knowledge about the species’ conservation status.

Materials & Methods

Occurrence data

In our analyses we used 625 lowland tapir location points, 500 for modeling (Table S1) and an independent 125 for testing (validating) (Table S2) the generated distributions. Location data were obtained from (Brooks, Bodmer & Matola, 1997; Anderson, 1997; Simonetti & Huareco, 1999; Patterson et al., 2003; Florez et al., 2008), and a data set developed from expert consultation and our own fieldwork.

Environmental descriptors

We used eight (8) environmental variables (0.04° of spatial resolution, ∼5 km) of which 6 climatic variables of WorldClim (Hijmans et al., 2005), as well as altitude and vegetation index (Table 1). These variables are commonly used in predictive species distribution, and represent a set of easily interpreted ecological variables.

Table 1 Environmental Variables (EV) used to model the potential distribution of Tapirus terrestris in South America.

All variables were resampled from original resolution to 0.04°(∼5 km), using the average value of all involved pixels, where the source pixels are covered by the target pixel.

Environmental Variable (EV)	Acronym	WorldClim acronym	Source	
Annual Mean Temperature	AMT	BIO1	WorldClim (Hijmans et al., 2005)	
Mean Temperature of Warmest Quarter	MTWQ	BIO10	
Mean Temperature of Coldest Quarter	MTCQ	BIO11	
Annual Precipitation	AP	BIO12	
Precipitation of Wettest Quarter	PWQ	BIO16	
Precipitation of Driest Quarter	PDQ	BIO17	
Altitude—Digital Elevation Model	ALT	–	Shuttle Radar Topography Mission (http://www2.jpl.nasa.gov/srtm/)	
MODIS Normalized Difference Vegetation Index (NDVI)-32 day composites-Oct/15–Nov/15/2004. Date of the composite represents well the contrast between forest and open formations.	NDVI	–	Global Land Cover Facility (GLCF) (http://www.landcover.org/data/modis/)	

Distribution models

We used ENFA version BioMapper 4.0 (Hirzel, Hausser & Perrin, 2007) and MAXENT version 3.2.3a (Phillips, Anderson & Schapire, 2006) models to describe habitat suitability and potential tapir distributions. Both methods use environmental data linked to species location points and relate this to environmental variables across the area of interest. For the T. terrestris Consensual Habitat Suitability Map (CHSM) the simple average of all models outputs was calculated. For the T. terrestris potential distribution binary map (suitable/unsuitable), we applied the Minimum Training Presence (MTP) as a threshold value for models and CHSM, because it is the most conservative threshold, identifying the maximum predicted area possible while still maintaining a zero omission rate for both training and test data. Darren (2014) identifies MTP with more appropriate threshold criteria for T. terrestris, based on its own broad distribution and variety of habitats used by the species.

Additionally, for comparative purposes, the images resulting from each of the ENFA and MAXENT models (with continuous values from 0 to 1) were reclassified into five environmental suitability zones, (1) an Unsuitable Zone (UNSZ; value pixel suitability < Minimum Training Presence, MTP), (2) a Low Suitability Zone (LSZ, value pixel suitability between MTP value and 0.25), (3) an Intermediate Suitability Zone (ISZ, value pixel suitability between 0.25 and 0.50), (4) a High Suitability Zone (HSZ, value pixel suitability 0.50 and 0.75), and (5) a Very High Suitability Zone (VHSZ, value pixel suitability > 0.75).

Ecological Niche Factor Analysis (ENFA)

The ENFA approach uses a factor analysis similar to Principal Component Analysis when producing species distributions (Hirzel et al., 2002). ENFA analyzes many environmental variables (EV) and reduces them to a few uncorrelated factors. This information is then used to produce an ecologically influenced species distribution. In ENFA all factors have ecological weight. The first factor is called Marginality (M), and measures the difference between the average conditions at sites where individuals of the species where actually located (species distribution) compared to sites throughout the entire area of interest (global distribution), to produce a distribution of the species’ niche in this environmental space. Another factor that is also considered is Specialization (S), which is the ratio of global variance to species variance. This item is a measure of niche breadth for the species (Braunisch et al., 2008). An M value close to one indicates that the species is a habitat specialist relative to the average condition of all EVs. The inverse of Specialization (1/S) is global Tolerance (T), which is a measure of the ecological flexibility of the species. A low value of T (close to 0) identifies a “specialist” species that tends to live in a very narrow range of conditions. A high value of T (close to 1) indicates a species that is not very selective of its living environment.

A Habitat Suitability Map (HSM) factor is calculated using the median—extremum algorithm derived from the first factors. This is the preferred algorithm for use when the real optimum is located at the extremes of the environmental conditions. We used broken-stick heuristics to determine the number of significant factors that should be retained to calculate habitat suitability (see, Jackson, 1993).

MAXENT

MAXENT uses a machine learning response to predict species distributions from incomplete data. This method estimates the most uniform distribution (maximum entropy) of the sampled points relative to background locations across the study area. It produces a model of a species’ environmental requirements based only on presence data and a set of environmental variables (Phillips, Anderson & Schapire, 2006).

MAXENT assumes that sampling of presence locations is unbiased. In MAXENT spatial biased sampling promotes model inaccuracy (Phillips, Anderson & Schapire, 2006; Phillips et al., 2009; Syfert, Smith & Coomes, 2013). To account for the spatial bias in presence records, we used the bias grid (Fig. S1), following procedures outlined by Elith, Kearney & Phillips (2010). The bias grid is used to down-weight the importance of presence records from areas with more intense sampling. The weighting surface is calculated based on the number of presence records within an area around any given cell (weighted by a Gaussian kernel with a standard deviation of 100 km).

MAXENT also provides environmental variable response curves indicating how each variable affects the predicted distribution. We ran MAXENT to model lowland tapir distribution under the ‘auto-features’ mode and the default settings with 10-fold replicates (jack-knife cross-validation). The logistic output was used (habitat suitability on a scale of 0–1), with higher values in the Habitat Suitability Map (HSM) representing more favorable conditions for the presence of the species (Elith et al., 2006; Phillips & Dudík, 2008).

Model validation and comparison

Although validation procedures based on resampling of input data have some merit in simulating species occurrence, they fail to provide the same degree of confidence as when using an independent dataset (Greaves, Mathieu & Seddon, 2006). Thus, to evaluate the predictive capacity of the models, two approaches were used: the first—Model Fit—tested the fit of occurrence points to the generated models; for ENFA using the Boyce index (B) with 10-fold jack-knife cross-validation (for more details, see Boyce et al., 2002; Hirzel et al., 2006). For the MAXENT model, we used 10-fold replicates (jack-knife cross-validation) to obtain the average Area Under Curve (AUC) of the Receiver Operating Characteristics (ROC) analysis. The second approach used was—Field Truth; this validation method used an independent set of 125 actual occurrence records (randomly selected from total points and not used in the generation of models) to evaluate the predictive capacity of the models. The predicted suitability of the models was extracted for each test point, and the average suitability was used to evaluate the model accuracy.

We compared the generated ENFA and MAXENT lowland tapir models using Fuzzy index for continuous maps, and Kappa index for potential distribution binary maps (suitable/unsuitable through MTP threshold criteria) using the Map Comparison Kit v.3.2 software developed by the Netherlands Environmental Assessment Agency (Visser & Nijs, 2006). Both indices express the pixel similarity for a value between 0 (fully distinct) and 1 (fully identical).

Additionally we used Olson et al.’s (2001) delineation of the terrestrial “Ecoregions of the World” as our base map (Fig. 1) to better demonstrate the comparison between models and to quantify habitat loss in a South American ecoregions context.

Figure 1 Terrestrial Ecoregions (adapted from Olson et al., 2001) and locations of lowland tapir (Tapirus terrestris) occurrence in South America.

Potential distributions versus remaining natural vegetation and protected areas

In order to identify both habitat availability and how effective the existing protected areas network is for T. terrestris, a Consensual Potential Distribution Map (CPDM, derived from CHSM—Consensual Habitat Suitability Map—reclassified as suitable and unsuitable, based on MTP cutoff criteria), was overlaid with the Land Cover Map for South America (Eva et al., 2002), upgraded for Brazil (MMA, 2009), and with the WDPA map of protected areas (WDPA, 2014). For these analyses the Land Cover Map for South America was reclassified as Anthropic, Grassland and Forest classes and the protected areas network was subdivided into two categories: Strict Protection (IUCN Categories I, II, III and IV) and Sustainable Use areas (IUCN Categories V, VI and Indigenous Territories identified in WPDA map).

Results

Lowland tapir distribution with ENFA

The ENFA model explained 85.5% of the information (100% of the Marginality and 71% of the Specialization) based on the two factors selected by the broken-stick heuristics criterion for extrapolating lowland tapir distributions (Fig. 2A). Cross-validation of the model quality resulted in a Boyce index of 0.62 ± 0.14, indicating a satisfactory predictive capacity (model fit). Analysis of the average suitability of test records using Field Truth produced a value of 55.48 (SD 28.15), indicating high accuracy for the model, since this average value corresponds to the High Suitability Zone for the species. Figure 2B represents the ENFA potential distribution binary map (suitable/unsuitable) based on the Minimum Training Presence cutoff criteria (MTP = 0.02).

Figure 2 (A) ENFA Habitat Suitability Map; (B) ENFA potential distribution binary map (suitable/unsuitable) based on the Minimum Training Presence cutoff criteria (MTP = 0.02); (C) MAXENT Habitat Suitability Map; (D) MAXENT potential distribution binary map (suitable/unsuitable) based on the MTP cutoff criteria (MTP = 0.08).

Unsuitability Zone (UNSZ), Low Suitability Zone (LSZ), Intermediate Suitability Zone (ISZ), High Suitability Zone (HSZ), and Very High Suitability Zone (VHSZ) identified.

An overall M value of 0.57 and T of 0.52, indicates that lowland tapir habitat differs moderately from the average conditions across the entire distribution area, suggesting the species is moderately tolerant of a range of conditions. The M factor alone accounted for 35% of the total specialization, indicating an intermediate niche breadth for lowland tapirs (see Hirzel et al., 2004).

The relative contribution of EV to the ENFA marginality factor (Fig. 3A) indicates that lowland tapirs “prefer” (more suitability) warm-humid areas with dense forest cover (Annual Mean Temperature between 21 °C and 27 °C; Mean Temperature of Warmest Quarter between 23 °C and 28 °C; Mean Temperature of Coldest Quarter between 18 °C and 25 °C; Annual Precipitation of 1,076–2,654 mm; Precipitation of Wettest Quarter of 485–1,023 mm; higher values of NDVI) and avoid high altitude areas. The highest specialization for the species (Fig. 3A) was associated with the temperature variables (Annual Mean Temperature, Mean Temperature of Warmest Quarter, Mean Temperature of Coldest Quarter, respectively), showing some sensitivity (low tolerance) to shifts away from their optimal values on these variables.

Figure 3 (A) The relative contribution of Environmental Variables (EV) to the ENFA Marginality and Specialization factors—EVs are sorted by decreasing absolute value of coefficients on the marginality factor. Positive values on this factor mean that T. terrestris prefers locations with higher values on the corresponding EV than the average value in the study area. Signs of coefficient have no meaning for the specialization factors. (B) Jackknife test results of individual environmental variable importance in the development of the MAXENT model relative to all environmental variables (hactched bar), for each predictor variable alone (black bars), and the drop in training gain when the variable is removed from the full model (gray bars).

An overlay of the ENFA-identified VHSZ and HSZ areas with Olson et al.’s (2001) delineation of the terrestrial ecoregions of the world shows that the best areas for lowland tapirs occur in Tropical Moist Broadleaf Forests (Figs. 1 and 2A). The Tropical Moist Broadleaf Forests of the northern Brazilian Amazon, southern Venezuela and the lowlands of Colombia and Peru, northern Cochabamba and southern Beni Department of Bolivia where also identified as VHSZ areas for lowland tapirs. In contrast, areas south and east of Amazon River basin, the Llanos Savannas biome of Venezuela and Colombia, and the central and north Cerrado Biome (Brazil) were deemed as slightly less (HSZ) suitable for lowland tapirs. An ISZ was identified in the western portion of the Cerrado, the Pantanal Wetland, Atlantic Forests (mainly the coastal region), Chiquitano and Dry Forests regions. The least suitable (LSZ) vegetation types are southern subtropical grasslands, southwestern thorn scrub vegetation of the Dry Chaco biome and the eastern (west of Atlantic Forests) transition zone between Caatinga, Cerrado and Atlantic Forest regions of Brazil. These areas are dominated by tropical seasonal semi deciduous forests (Oliveira-Filho, Jarenkow & Rodal, 2006) and apparently delineate the distributional limit of lowland tapirs. A large part of the Caatinga biome was classified as unsuitable (UNSZ) for lowland tapirs, particularly the eastern half of this region.

Lowland tapir distribution with MAXENT

With an average AUC of 0.804 (SD = 0.01; 10-fold replicates), the MAXENT model (Fig. 2C) achieved a satisfactory model fit and the modeled distribution performed better than random. A Field Truth value of 51.13 (SD = 13.51) indicates that the model achieved high accuracy. This average value corresponds to the High Suitability Zone for lowland tapirs. Figure 2D represents the MAXENT potential distribution binary map (suitable/unsuitable) based on the MTP cutoff criteria (MTP = 0.08).

The Mean Temperature of the Coldest Quarter (MTCQ) was the variable with the highest gain and which most decreased gain when omitted (when used in isolation) from the model (Fig. 3B). The response curves (Fig. S2) for the EV of this model indicate that lowland tapirs are strongly associated with warmer regions (MTCQ between 15 °C and 23 °C, and AMT between 20 °C and 25 °C) and areas with an annual precipitation over 1,000 mm (suitability of presence > 0.5).

With MAXENT the VHSZ areas for lowland tapirs were very restricted to the Eastern Cordillera Real Montane forests in Ecuador. The slightly lower quality HSZ areas prevail in the northern Tropical Moist Broadleaf Forests biome of Colombia, Ecuador and Bolivia. This zone also predominates in Paraguay, northern Argentina, Atlantic Rainforest, the Pantanal Wetland, and the Chiquitano Dry Forests of Bolivia. The ISZ equaled the biggest area identified by MAXENT. The LSZ was found in the Caatinga Biome, in the subtropical highland grassland in the south of the Atlantic Rainforest Biome, and the southern range of its modelled distribution. Some parts of the Caatinga (areas surroundings the São Francisco River, Brazil) biome were classified as LSZ, but the region immediately to the west—a transition area between the Caatinga and Cerrado—supports relatively high values of suitability (ISZ).

Comparison of models and Consensual Habitat Suitability Map (CHSM)

The spatial similarity between HSMs produced by the ENFA and MAXENT was moderate, as indicated by the intermediate value of the Fuzzy (0.53). However, if the cutoff limit for suitability is MTP, the Kappa similarity value is very high (0.80) between the models, indicating a similar geographical range between predicted distributions.

In the CHSM (Fig. 4A) areas with higher habitat suitability values (VHSZ and HSZ) were identified in the Amazon region, Pantanal Wetland, Humid Chaco in Paraguay, and the Chiquitano Dry Forests of Bolivia. The Caatinga biome and the southern border of the modeled distribution correspond to areas with less habitat suitability in this map (LSZ). The MTP cutoff criteria (MTP = 0.06) was applied to this map (CHSM) to generate the Consensual Potential Distribution Map (CPDM) shown in Fig. 4B.

Figure 4 (A) Consensual Habitat Suitability Map, CHSM; (B) Consensual Potential Distribution Map, CPDM (suitable/unsuitable), based on the Minimum Training Presence cutoff criteria (MTP = 0.06).

Unsuitability Zone (UNSZ), Low Suitability Zone (LSZ), Intermediate Suitability Zone (ISZ), High Suitability Zone (HSZ), and Very High Suitability Zone (VHSZ) identified.

For a more conservative approach the overlap between the modeled area and the known Tapirella bairdii non sympatric distribution with T. terrestris, on the Pacific coast in Colombia and Ecuador (Brooks, Bodmer & Matola, 1997; Patterson et al., 2003; Schank et al., 2015), was withdrawn from the final map CPDM (for more details see the ‘Discussion’).

Potential distributions (CPDM) versus remaining natural vegetation and protected areas

The Consensual Potential Distribution Map (CPDM) covers 13,441,402 km2, of which 29.44% are anthropogenic, such that 9,484,379 km2 are available for the species (Table 2). The Atlantic Forests, Chocó Darién Moist Forests, Caatinga biome and Tropical and Subtropical Dry Broadleaf Forests (extreme north of South America) are the ecoregions with the largest individual habitat losses (Table 3). However, considering the size of the lost area (in km2), the Cerrado, Atlantic Forest and Amazon Region (Tropical and Subtropical Moist Broadleaf Forests) presented the largest losses. The Amazon region represents 62.73% (5,949,846 km2) of the total (9,484,379 km2) suitable and remaining area for T. terrestris.

Table 2 Land Cover (remaining vegetation) and protected area network in modeled Tapirus terrestris potential distribution (Consensual Potential Distribution Map, CPDM).

Land cover class	Areaa (km2)	Area within a Strict Protection protected areaa (km2)	Area within a Sustainable Use protected areaa (km2)	Protected Areas network extenta (km2)	
Forest	7,003,896 (52.11%)	690,277 (81.37%)	1,927,908 (82.70%)	2,618,185	
Grassland	2,321,326 (17.27%)	114,816 (13.54%)	219,451 (9.41%)	334,267	
Water	159,157 (1.18%)	8,351 (0.98%)	18,831 (0.81%)	27,182	
Anthropic	3,957,023 (29.44%)	34,834 (4.11%)	165,105 (7.08%)	199,939	
Total (km2)	13,441,402	848,278 (6.31%)	2,331,295 (17.34%)	3,179,573 (23.66%)	
Notes.

a Values within parenthesis indicate its percentage.

Table 3 South American Ecoregions (adapted from Olson et al., 2001), anthropic and remaining natural areas in modeled Tapirus terrestris potential distribution (Consensual Potential Distribution Map, CPDM).

Ecoregions	Anthropica (km2)	Remaina (km2)	Total (km2)	
Amazon Region—Tropical and Subtropical Moist Broadleaf Forests	846,274 (12.45)	5,949,846 (87.55)	6,796,120	
Atlantic Forests	939,594 (80.46)	228,205 (19.54)	1,167,799	
Caatinga Brazilian Biome	478,964 (66.16)	244,964 (33.84)	723,928	
Cerrado Woodlands and Savannas	967,354 (51.15)	923,911 (48.85)	1,891,265	
Chiquitano Dry Forests	51,120 (23.58)	165,718 (76.42)	216,838	
Chocó Darién Moist Forests	55,401 (69.96)	23,794 (30.04)	79,195	
Deserts and Xeric Shrublands	48,042 (35.72)	86,460 (64.28)	134,502	
Dry Chaco	106,582 (15.77)	569,329 (84.23)	675,911	
Flooded Grasslands and Savannas	5,398 (9.76)	49,905 (90.24)	55,303	
Humid Chaco	43,822 (15.23)	243,950 (84.77)	287,772	
Llanos Savannas	56,034 (13.87)	347,900 (86.13)	403,934	
Mangroves	14,467 (31.22)	31,874 (68.78)	46,341	
Montane Grasslands and Shrublands	271 (5.12)	5,024 (94.88)	5,295	
Pantanal Flooded Savannas	25,081 (15.55)	136,238 (84.45)	161,319	
Temperate Grasslands, Savannas, and Shrublands	31,516 (36.57)	54,672 (63.43)	86,188	
Tropical and Subtropical Dry Broadleaf Forests	123,732 (54.72)	102,375 (45.28)	226,107	
Tropical and Subtropical Grasslands, Savannas, and Shrublands	163,371 (33.78)	320,214 (66.22)	483,585	
Total	3,957,023 (29.44)	9,484,379 (70.56)	13,441,402	
Notes.

a Values within parenthesis indicate its percentage. Adapted from Eva et al. (2002), and upgraded for Brazil by MMA (2009).

In this context, the protected areas network covers/protects 23.66% (3.179.573 km2) of the total suitable area for T. terrestris, as follows: 848,278 km2 Strict Protection and 2,331,295 km2 Sustainable Use. Only 6% of the remaining Cerrado area suitable for lowland tapir is within a Strict Protection protected area. For the Atlantic Forest and Amazon region the remaining area under strict protection is 10%.

Discussion

Our study presents a distribution and habitat map for lowland tapir in South America. We also describe the potential habitat suitability of various geographical regions, habitat loss and assessment of the effectiveness of a protected areas network. Additionally, we evaluated the predictive capacity of two modeling approaches for describing these patterns.

While the environmental requirements identified by the ENFA and MAXENT-modeling approaches for describing lowland tapir range appears broadly similar, only the ENFA model identified forest cover density (NDVI) as a factor contributing to tapir habitat suitability. This resulted in ENFA identifying the Amazon Region as a VHZ or HSZ for lowland tapirs (overlay of Figs. 1 and 2A). This result is supported by field knowledge on the ecology of this species, where tapirs have been identified as strongly associated with warm and wet regions (Bodmer, 1991; Fragoso, 1997; Tobler, 2008; Taber et al., 2009).

In contrast, MAXENT identified much of the Amazon Region as an area of lower suitability for tapirs (ISZ; Fig. 2C). This result, in spite of using the bias grid, is related to an idiosyncrasy of the technique, in that MAXENT establishes a complex (very parameterized) and strong fit (over fit) between dependent and independent variables (Jiménez-valverde, Lobo & Hortal, 2008; Kamino, Stehmann & Amaral, 2011; Rangel & Loyola, 2012). This explains why the relatively low number of tapir records in the very large Amazon region led MAXENT to identify the region as a lower suitability zone for lowland tapirs. In contrast, results from areas at the climatic extreme of tapir tolerance, such as the xeric Central Chaco, where more records were available, where identified counter intuitively (based on ecological field information) by MAXENT as highly suitable for tapirs. This classification reflects a bias in the distribution pattern of occurrence records that is related to the difficulty of conducting research in the vast, remote Amazon region (Brooks, Bodmer & Matola, 1997) relative to more easily accessed, spatially restricted biomes, rather than to the real suitability of areas of lowland broadleaf forests for lowland tapirs.

Both models identified the Chocó-Darién Moist Forests ecoregion (western end of Colombia and Ecuador) as suitable for the lowland tapir (Figs. 1 and 2). This region is also the known South American range limit for the Central American Baird’s tapir. This potential area of overlap for the two tapir species occurs because of the environmental similarity of this ecoregion (within the context of EV used) with adjacent areas-such as the Magdalena-Urabá moist forests—which contain records of lowland tapirs and form a continuous corridor with the lowland forests of the western Andes up to a bottleneck region between the Pacific ocean and the western slope of the Andes in southeastern Ecuador. The presence or absence of either tapir species in this region may be partially related to interspecific interaction between the species. The models in the context of EV used did not detect this possibility. This aspect (limitation) of both models, combined with the already described T. terrestris distribution, were the main reasons for excluding this region from the potential distribution map (CPDM) for the analyses of remaining habitats availability and effectiveness of the protected areas network.

Conclusions

Apparently viable tapir populations in the protected areas of eastern Brazil (Medici, 2010; Eduardo, Nunes & Brito, 2012) were classified as falling into LSZ, ISZ or HSZ, depending on the modeling method used. Tapir population levels here are low and this information is linked to the forest types by the models. However, low population levels here are likely the result of human activities that have decreased tapir densities, such as hunting and habitat destruction, rather than environmental factors (Taber et al., 2009). That is, the forests of eastern Brazil and their transition zones to the seasonal forests of the adjacent Caatinga and Cerrado regions of eastern Brazil have had their tapir populations reduced or extirpated by anthropogenic impacts, so that low population sizes are now associated with these ecosystems and are interpreted by the model, which does not separate anthropogenic variables from non-anthropogenic variables, as being correlated with the ecosystem.

In this context, our results indicate that Brazilian ecoregions have the highest habitat loss for the tapir, which supports the results obtained by Taber et al. (2009) and Medici et al. (2012). Cerrado and Atlantic Forest account for nearly half (48.19%) of the total area lost (1,906,948 of 3,957,023 km2).

When associated to the well-known hunting pressure and elevated habitat loss for the Caatinga, our low habitat suitability results for this biome support the hypothesis of a probable local extinction of tapir indicated by Taber et al. (2009). The same logic can be applied to the southern limit of the tapir distribution area within the Pampa region (Temperate, Tropical and Subtropical Grassland, Savannas, and Shrublands Ecoregions).

The Amazon region contains the largest extent of land under protection, and the most extensive remaining habitat for the tapir, but also showed high levels of habitat loss outside protected areas. This increases the importance of adequate monitoring of protected areas, so as to determine the relative effectiveness of indigenous territories, strict protection areas and sustainable use areas in sustaining tapir populations and inform the management of these areas. Management and use by humans is an inherent characteristic of an area; once the impact of management category on tapir populations is understood, this information can be added to habitat suitability models.

In conclusion, MAXENT and ENFA produced different HSM for the lowland tapir. While MAXENT was efficient at identifying areas as suitable or unsuitable, it was less efficient (when compared to the results by ENFA) at identifying the gradient of habitat suitability. MAXENT is a more multifaceted technique that establishes more complex relationships between dependent and independent variables. It is an excellent tool for describing spatial occurrence data; however, spatial aggregation of occurrence records can lead to the miss-classification of areas as highly suitable when they are not, and the identification of areas that are highly suitable as exhibiting poor or no suitability for the species. As conservation planners and ecologists we should remember the axiom that “…all models are wrong, the practical question is how wrong do they have to be before they are not useful” (Box & Draper, 1987).

If the objective of a conservation or research program is to identify areas that are environmentally very similar to the points where species have been noted, without concern for understanding the ecological and human factors that contribute to that occurrence, then MAXENT is well suited for the task. However, our results indicate that ENFA is more appropriate for the task of classifying habitat suitability zones and species distribution patterns, not only because of the accuracy of the generated models but also due to this method’s ability to better identify the gradient of habitat suitability across the potential distribution range, rooted in solid and clear (easy interpretation of parameters) ecological theory (Rangel & Loyola, 2012).

All tapir species are considered as being at risk throughout their ranges (TSG-IUCN, 2015). While the lowland tapir still exhibits robust populations in much of its extensive range, in other very large areas populations have become fragmented and highly threatened. Conservation planning for the four species, especially those that are listed in red data books, requires the use of the most robust methods for determining potential population size, abundance patterns, distribution and factors influencing these variables. Our results demonstrate that for at least one species, the lowland tapir, the use of a consensual approach better reflected its current distribution patterns, confirming the critical situation of this species in Brazilian ecoregions.

Given that many governments and NGOs now use modeling techniques to assess species habitat suitability zones and distribution patterns for conservation planning, we strongly recommend that care be taken to select the most appropriate model.

Supplemental Information

Table S1 Lowland Tapir location points (n = 500) used for modeling (Raw Data)

Click here for additional data file.

Table S2 Lowland Tapir location points (n = 125) used for test model (Raw Data)

Click here for additional data file.

Figure S1 MAXENT Bias Grid

MAXENT bias grid, according procedures outlined by Elith, Kearney & Phillips (2010). The bias grid was used to down-weight the importance of presence records from areas with more intense sampling. The weighting surface was calculated based on the number of presence records within an area around any given cell (weighted by a Gaussian kernel with a standard deviation of 100 km).

Click here for additional data file.

Figure S2 Response-curves MAXENT distribution model

Response-curves of the variables in the MAXENT Tapirus terrestris distribution model. Mean Temperature of Coldest Quarter (MTCQ); Annual Mean Temperature (AMT); Annual Precipitation (AP). These curves show how each environmental variable affects the MAXENT prediction when all environmental variables are used to build the model.

Click here for additional data file.

The authors thank the IUCN/SSC Tapir Specialist Group for valuable discussions, especially Patrícia Medici. We thank Andressa Gatti, Ana P Carmignotto, M de Vivo, JA de Oliveira, T de Oliveira, A Vogliotti, FHG Rodrigues, D Sana, PG Crawshaw, RV Rossi, A Gomes Filho, FL Sicuro, C Leôncio, AR Percequillo, AMR Bezerra, AC Borges, Hipólito Neto, Manoel dos S Filho, M Mazzolli, P Rodrigues Gonçalves, PP do Amaral, RM Falleiro, M Jardim, M Pérsio, MR Barroeta, L Gondim, Sérgio M Vaz, FR Tortato, MA Tortato, T Bagatini, Mario MR Júnior, and L Tiepolo and others for contributing data. Andrew Taber, Silvia Eric Sanderson, C Chalukian, Karen Minkowski, Damián Rumiz, Eduardo Ventincinque, Edsel Amorim Moraes, Jr, and Andrew Noss, provided support in the gathering of information on species distribution.

Additional Information and Declarations

Competing Interests

Author Contributions

Data Availability

The authors declare there are no competing interests.

Jose Luis Passos Cordeiro and Luiz Flamarion B. Oliveira conceived and designed the experiments, performed the experiments, analyzed the data, contributed reagents/materials/analysis tools, wrote the paper, prepared figures and/or tables, reviewed drafts of the paper.

José M.V. Fragoso conceived and designed the experiments, analyzed the data, contributed reagents/materials/analysis tools, wrote the paper, reviewed drafts of the paper.

Danielle Crawshaw performed the experiments, analyzed the data, contributed reagents/materials/analysis tools, wrote the paper, reviewed drafts of the paper.

The following information was supplied regarding data availability:

The raw data has been supplied as Supplemental Information.

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
