# Peer review of "Lowland tapir distribution and habitat loss in South America"

_PeerJ, doi:10.7717/peerj.2456_

## Round 0.1 · original submission · Major Revisions

Both reviewers fell the paper is not well structured so it is hard to follow. It is necessary to clarify the main questions or hypoththeses and complete relevant information throught the paper .I also suggest to have a native English speaker review your manuscript or hire an editing service/editor to edit it, before resubmit it the for publication.

Additional and necessary useful changes are suggested by both Reviewers. I strongly recommend to follow them carefully.

·

Basic reporting

This paper investigates the potential species distribution of lowland tapir in South America. To do so, the authors used two species distribution modeling techniques, in conjunction with a training and independent test data set of know species occurrences and a set of environmental variables.

Unfortunately, I felt that the paper suffers from several shortcomings. First, it is not very well structured, and quite hard to follow. This is especially true for the Introduction, which starts with basic information on the ecology and natural history of lowland tapir, rather than with the topic at hand. As a matter of fact, throughout the manuscript it remains unclear what the main gist of the paper is; is it SDM of lowland tapir, or recommendations for conservation, or a comparison of species distribution methods, or the value of SDM model selection for conservation? I guess and recommend it would be SDM of lowland tapir for conservation purposes, but this should then be clearly stated, and the Introduction should be restructured accordingly. While much (but not all!) information is provided, the reader is required to filter out the main question/hypotheses. While PeerJ does not take into account the potential impact of the paper, I do think the authors should state how their paper will contribute to a better understanding of their topic of choice, which remains unclear (again, I can guess, but there is no information regarding for instance the conservation status of lowland tapir or threats to their long term survival).

The Discussion is very focused on a comparison of methods. I think it should focus more on the implications of the results for the conservation of tapir (what I guess would be the gist of the paper). I missed recommendations resulting from this study. Further, the Discussion is quite repetitive, and should be condensed (but recommendations and a discussion on the distribution of lowland tapir should be added).

Further, the paper omits relevant information in some places – I have indicated specific cases in other sections. As for proper language usage, the manuscript is overall clear, but does need to be edited to make sure correct English is used; wording and grammar is not always correct (one thing that is prevailing throughout the ms, is that articles are often omitted).

Experimental design

Evaluating the experimental design of this paper was not straightforward, because the gist/main question remains unclear. However, for a method comparison (as one might conclude from the discussion), the paper is not technical enough and uses too few different SDM approaches, so I assumed a focus on SDMs of lowland tapir for conservation purposes. Although the input data may be appropriate, too little information is provided to evaluate whether this is the case, or how data were treated. The most glaring omission is information on the environmental variables. Were there 9 variables used (line 100), or 8 (the number I counted in Table 1)? Why were these variables picked over others? Based on prior knowledge about lowland tapir ecology / natural history, what is their supposed relevance to tapir distribution? Are there any cross-correlations between variables (please provide a table), and did the authors use a threshold above which variables were dropped? The climate data represent 50-year averages, but why was only 1 month from 1 year of NDVI data used, instead of multi-year averages? It seems that one should account for inter-annual variation in NDVI. How were environmental variables resampled to get at the 0.04 degree resolution? In Table 1 this is stated only for SRTM, but I assume this was also done for the other variables?

As for the species records, I appreciate the value of an independent data set. However, how is the way the authors did this different from dividing the full data set up into training and test sites, except that the final full models provided by the authors do not include the test sites? I would strongly recommend running many (100s – 1000s) models with randomly selected training versus test sites.

It is great to see that the authors recognize the caveats of species presence records in this area, and generated a bias grid to account for observation bias. I am not familiar with ENFA, but it seems that the bias grid was only used in Maxent, and not in ENFA. The statement in lines 148-150 is not only true for Maxent, but also for ENFA – and as a matter of fact for all SDM approaches. I am left wondering whether the observed differences in species distributions between Maxent and ENFA models cannot be boiled down to the former accounting for observation bias, and the latter not. It would help generating Maxent models without bias grid to further evaluate the origin of the differences between models.

What is the rationale for using Maxent and ENFA? Is there any suggestion that they are complementary? If the idea is to evaluate the performance of different models or to come up with the best possible consensus map, one should include many more models, such as is made possible by biomod or biomod2.

The authors use the Minimum Training Presence as a threshold to convert continuous output maps into binary presence/absence maps. The basis for this is not well supported, and it is likely that habitat suitability is vastly overestimated. Using the MTP assumes that all species records were in suitable habitat, and weighs each record equally. However, in some areas the species may have been observed while dispersing from one area of high suitability to another area of high suitability through and area of low suitability (but good enough to cross). There is no analysis of outlier habitat conditions, so it is hard to evaluate to what extent this may be a problem. A sensitivity analysis of the influence of the threshold used would be very welcome here.

In addition to the above, the authors use the MTP as a threshold for identifying unsuitable areas, and values of 0.25, 0.5, and 0.75 for the remaining zones of suitability. Why such arbitrary values? Is there any rationale to this, or would it be better to use more informative thresholds, such as balancing threshold, or percentiles?

It remains unclear how models were tested using the ‘field truth’ method. Was the predicted suitability (line 173) converted to presence/absence using the minimum training presence threshold? More information is needed. Again, the procedure followed might be fine, but it’s difficult to evaluate with the amount of information provided.

Finally, I am not convinced that the procedure of the ‘fuzzy model comparison’ is quite correct. How should the values of ~ 0.5 be interpreted? The problem is that there appears to be a big difference in the number of cells within the 5 zones of suitability. If most cells are within 1 or two categories (i.e. zones), two maps with a completely random assignment of values to grid cells, but with the same number of cells in each category are expected to be pretty similar (for instance, if 50% of the grid cells are within category 1, and the other 50% in category 2, then 50% of the grid cells are expected to be exactly the same between two random models! If 50% of the grid cells are within category 1, and the other 50% in many other categories, then still ~25% of the grid cells are expected to be exactly the same between two random models.). For correct interpretation of these results, there really needs to be an evaluation against a random expectation.

Validity of the findings

I am afraid that I cannot quite agree with the statement in line 288 that this “study presents a rigorously derived distribution and habitat map for lowland tapir”. I appreciate the difficulty of acquiring good species records in this area, and I acknowledge the authors’ attempts to correct for observation bias. However, it would have been great if bias was also accounted for in the ENFA models. We are, however, left wondering whether the differences between models is due to observation bias in ENFA. As for the environmental variables, it is great that the authors included both climate and vegetation variables, but we need much more information on their selection criteria and variable processing procedure.

The Discussion and Conclusion focus mostly on a comparison between models that is not well supported by the results. For instance, what is the basis for stating that Maxent is incorrectly identifying the Amazon region as an area of lower suitability, related to overfitting (lines 299-302)? This may be true, but first: why do you think the Maxent model is incorrect in doing so, whereas the ENFA model is performing better in that respect? The ‘field truth’ values are very similar between the models, so what is the basis of disregarding one result over the other? Second, if Maxent is indeed performing worse than ENFA, I did not see an evaluation of the potential cause of this. Again, the explanation provided may be true, but please assert caution and make sure that it is clear that you are speculating here. Further, in the same discussion the observation bias is brought up as a cause of the purported incorrect Maxent model, even though Maxent was the only method in which the bias grid was used. I would thus expect that ENFA would be much more biased than Maxent. Similarly, I don’t see any support for the statement that Maxent was less efficient in identifying that gradient of habitat suitability (line 356). The statement that follows (lines 361-363) is just as well applicable to the ENFA method, so seems out of place here.

As for the exclusion of areas where Baird’s tapir occurs, what is the support that they are competitors and exclude one another from that area? Do you have references? If so, please provide.

The statement in lines 341-345 seems odd to me. If local absence in this region is caused by extirpation due to hunting, the model should predict the area to be suitable. It cannot predict habitat unsuitability and with that support local extirpation due to hunting, unless this area is characterized by very specific habitat conditions that coincide with high hunting pressure. Please be careful in making such shortcuts and blunt statements.

The conclusion that “the use of a consensual approach better reflected its current distribution” (line 378) is not supported by the results – nor is it falsified; it has simply not been evaluated. We see two different species distribution maps resulting from two different methods, based on the same input data. Both methods perform well based on the associated evaluations, and for both methods the ‘field truth’ results are similar. Thus, we see two different maps, but based on the analyses done, we can’t evaluate which one is better.

The final sentence of the conclusion is a vague statement that I second, but its value was hardly demonstrated in this paper. If at all, the take-home message would be that one can come up with two different maps that appear to be based on two equally performing methods, yet imply different conservation strategies.

Additional comments

Line 112 – it really depends on how one defines conservative; it retains all training data, but likely overestimates the area of suitable habitat

Line 202 – that seems a high sd. As a result I don’t necessarily think the field truth value indicates high model accuracy.

Line 219-234 – All fine, but this stresses the need for a more rigorous discussion on how this study contributes to what we already know about tapir ecology; it seems hardly a surprise that they are a forest species.

Line 275 – the fact that ~30% of the CPDM covers anthropogenic habitat is not very convincing in terms of model performance and the appropriateness of the environmental variables. It seems that habitat conditions not captured by these environmental variables should be included.

·

Basic reporting

In this paper they are applying two SDMs (ENFA and Maxent) to map the potential distribution of a tapir species and they describe the so-obtained maps. I find that this work would be much more interesting if they could add any biological hypothesis, or any conservation problem to test. I really missed an hypothesis or a problem to solve with their methods. What is the problem with this tapir? Why do we need to map its potential distribution? What they aim to test? Is this a conservation paper? (then I miss a more focus discussion about the conservation status of the tapir, conservation policies, implications of the obtained results, etc.) They might need to set some goals, different from “just” obtaining a map.
Also, the discussion is very short and the conclusions are very long. I would suggest the authors to move part of the conclusions to the discussion, and keep the conclusion as a short list of their highlights.
I think that this paper will be ready for publication after some changes in the structure of the paper and some in methodological steps of the maxent model (see below).
I hope that the authors will find this comments valuable for improving their work,
All the best,
Sara Varela

Experimental design

I would suggest some improvements for the maxent model (see below)

Validity of the findings

Their maps are ok. However, again, I missed an hypothesis.

Additional comments

Detailed comments:
Line 17. I would eliminate “Rigorously derived” from the sentence: a “Rigorously derived” potential distribution… I think it is better if you go to the point and you do not add judgements of the quality of your own analysis.

Line 86-89. The explanation of ENFA seems confuse here (but it is perfect in the method section…).
Line 89-90. This explanation of maxent is very poor. You might say something like “maxent output is closer to a statistic regression”
See: http://onlinelibrary.wiley.com/doi/10.1111/j.1541-0420.2012.01824.x/abstract
http://onlinelibrary.wiley.com/doi/10.1111/j.1472-4642.2010.00725.x/abstract

Line 91. I would finish the intro by summarizing the scientific goals of your study, not your methodology. Something like “we aim to map the climatically suitable areas for the species and compare it to the current distribution of the Tapir…because blabla” (what are the problems? Which are your working hypothesis?)
Line 100. Please, specify that you are using variables from the WorldClim model, and add Hijmans 2005 reference here, and maybe also the link to the webpage. It is important to clearly explain which variables you used for the analysis. Also, in table 1 and in the text, I would change your own created acronyms and use the worldclim acronyms for the worldclim variables (Bio1, Bio12, etc.).
Line 101. The word “traits” could be misleading. It is normally used to talk about species characteristics (e.g. body size, clutch size… etc.). I would not use it here.

Line 100. It seems (table 1) that you are using 8 variables, not 9. 6 climatic variables plus altitude and vegetation.

Also, it would be interesting to add here the name of the variables and why you decided to use these ones and not others, in the material section. The selection of the variables in an important step in the construction of your SDM. It would be nice if you can relate the selected variables with the species observed requirements.

Line 112. This sentence is wrong. The minimum training presence threshold will give you an overestimation of the potential range of the species. You have 0 omission, as you said, but a high commission rate (0s predicted as 1s). This means that the map is not going to be the “minimum”, but rather a “maximum” range.
You can see a test with virtual species here: maxent map with the minimum training presence highly overestimates the species’ range (it is in Spanish but I hope you can read it) http://www.revistaecosistemas.net/index.php/ecosistemas/article/view/831

Lines 116-120. The selected thresholds for reclassify the continuous map are arbitrary. Those “suitability” values are dependent on the prevalence (related to the ratio “number of background points” vs “number of occurrences”). Thus, 0.5 does not mean anything. And certainly, does not mean 50% of probability of finding the species. If your training area is big, and you selected a high number of points as a background sample, your suitability values will be low, and you might have very few pixels with more than 0.5 suitability values.
I would adjust this to your species, e.g., to the percentile distribution of the suitability for your sampled occurrences… something that might mean something for your species. Not arbitrary values.

Line 122-141. Very nice explanation of ENFA.
Line 150-153. But see http://onlinelibrary.wiley.com/doi/10.1111/j.1600-0587.2013.00441.x/abstract
Geographic filtering might not always improve the output of maxent.

Line 155. It would be important to specify the parametrization of your maxent model. I assume that you are using the default parameters, but even so, please, write it down.
In maxent you can tune the way the variables are entered in the model (hinge, exponential, etc.), and the regularization parameter (for penalizing complex models). Both parameters change the model output, so they should be specified (see below)
http://onlinelibrary.wiley.com/doi/10.1111/jbi.12227/abstract

Line 182-184. I do not understand how they use this map and why. You are just cutting the range maps to quantify the amount of potential area for the tapir in each different type of environment? I would add this GIS layer to materials, to explain the map first, and also add some information in the introduction, to help the readers to understand what it is for. This might be an objective of your study “quantify which kind of habitat (biome, whatever…) has the largest suitable area for this species…”

Line 288. Please, eliminate “rigorously derived”.
Line 299-303. This is because you used the default parameters, which means that you are not penalizing complex models (regularization parameter=1), which means that your final model might be too complex (overfit) and might underestimate the potential range of your species.
Lines 305-308. This is because maxent can fit lines, it is similar to a regression technique. This means that, based on your results, it might be fitting a line with temperature, or precipitation, “more temperature more suitable”, or “less precipitation more suitable”. Maxent might not have boundaries if you allow to fit lines, and thus, you might have extreme environments predicted as present. This is why setting the parameters correctly is so important.
Lines 326-328. Of course, both models have different scales, so, using the same thresholds for cutting both maps would generate different outputs. You might need to adjust the thresholds independently for each model, according to the distribution of the suitability in your occurrence points (quartiles).

Lines 355-356. How did you test that ENFA was better predicting the gradient of habitat suitability? Did you have abundance records? It is not clear for me if this is an expert opinion given after seeing the maps, or if you did have any data to test this issue.
see: http://onlinelibrary.wiley.com/doi/10.1111/j.1472-4642.2012.00892.x/full

---

## Round 0.2 · accepted · Accept

After evaluating your re-submission and your responses to the Reviewers and my comments on the previous version, I am pleased to tell you that I arrived at the decision that all issues raised in the peer review process are now settled and your revised ms is ready for being published.